# Derivatives of Stochastic Gradient Descent in parametric optimization

**Franck Iutzeler**
Université Paul Sabatier,
Institut de Mathématiques de
Toulouse, France.

**Edouard Pauwels**
Toulouse School of Economics,
Université Toulouse Capitole,
Toulouse, France.

**Samuel Vaiter**
CNRS &
Université Côte d'Azur,
Laboratoire J. A. Dieudonné.
Nice, France.

## Abstract

We consider stochastic optimization problems where the objective depends on some parameter, as commonly found in hyperparameter optimization for instance. We investigate the behavior of the derivatives of the iterates of Stochastic Gradient Descent (SGD) with respect to that parameter and show that they are driven by an inexact SGD recursion on a different objective function, perturbed by the convergence of the original SGD. This enables us to establish that the derivatives of SGD converge to the derivative of the solution mapping in terms of mean squared error whenever the objective is strongly convex. Specifically, we demonstrate that with constant step-sizes, these derivatives stabilize within a noise ball centered at the solution derivative, and that with vanishing step-sizes they exhibit $O(\log(k)^2/k)$ convergence rates. Additionally, we prove exponential convergence in the interpolation regime. Our theoretical findings are illustrated by numerical experiments on synthetic tasks.

## 1 Introduction

The differentiation of iterative algorithms has been a subject of research since the 1990s (Gilbert, 1992; Christianson, 1994; Beck, 1994), and was succinctly described as "piggyback differentiation" by Griewank and Faure (2003). This idea has gained renewed interest within the machine learning community, particularly for applications such as hyperparameter optimization (Maclaurin et al., 2015; Franceschi et al., 2017), meta-learning (Finn et al., 2017; Rajeswaran et al., 2019), and learning discretization of total variation (Chambolle and Pock, 2021; Bogensperger et al., 2022). When applied to an optimization problem, an important theoretical concern is the convergence of the derivatives of iterates to the derivatives of the solution. Traditional guarantees focus on asymptotic convergence to the solution derivative, as described by the implicit function theorem (Gilbert, 1992; Christianson, 1994; Beck, 1994). This issue has inspired recent works for smooth optimization algorithms (Mehmood and Ochs, 2020, 2022), generic nonsmooth iterations (Bolte et al., 2022), and second-order methods (Bolte et al., 2023).

Convergence analysis of iterative processes have predominantly focused on deterministic algorithms such as the gradient descent. In this work, we extend these results in the context of strongly convex parametric optimization by studying the iterative differentiation of the Stochastic Gradient Descent (SGD) algorithm. Since the seminal work of Robbins and Monro (1951), SGD has been a workhorse of stochastic optimization and is extensively employed in training various machine learning models (Bottou et al., 2018; Gower et al., 2019). A critical aspect of our work is based on the fact that the sequence of iterative derivatives in this stochastic setting is itself a stochastic gradient sequence.

The goal of this work is to answer the following question:

38th Conference on Neural Information Processing Systems (NeurIPS 2024).

*What is the dynamics of the derivatives of the iterates of stochastic gradient descent in the context of minimization of parametric strongly convex functions?*

Our motivation for studying this question is twofold. First, while iterative differentiation through SGD sequences is possibly not the most efficient way to differentiate solutions of convex programs, it is very natural in the context of differentiable programming and has already been motivated and explored in the machine learning literature (Maclaurin et al., 2015; Pedregosa, 2016; Finn et al., 2017; Ji et al., 2022). Second, existing attempts to provide stochastic oracle based methods to differentiate through convex programming solutions require more intricate algorithmic schemes than the conceptually simple iterative differentiation of SGD. Despite its conceptual simplicity, the answer to this question is not direct in the first place due to the joint effect of noise on the iterate sequence and its derivatives.

**Contributions.** The strongly convex setting ensures that the solution mapping is single valued and differentiable under appropriate smoothness assumptions. In this setting, we prove in Theorem 2.2 the **convergence of the derivatives of the SGD recursion toward the derivative of the solution mapping**, in the sense of mean squared errors:
• We first provide a general result for non-increasing step-sizes converging to some $\eta \geq 0$ (covering constant step-sizes schedules), for which we prove that the derivatives of SGD eventually fluctuate in a ball centered at the solution derivative, of size proportional to $\sqrt{\eta}$.
• With vanishing steps, this result implies that the derivatives of SGD converge toward the solution derivatives, and we obtain $O(\log(k)^2/k)$ convergence rates for $O(1/k)$ step-size decay schedules.
• We also study the interpolation regime, for which we show that the derivatives converge exponentially fast toward the derivative of the solution mapping.
All these results suggest that derivatives of SGD sequences behave *qualitatively* similarly as the original SGD sequence under typical step size regimes.

The key insight in proving these results is to interpret the recursion describing **the derivatives of SGD as a perturbed SGD sequence**, or SGD with errors, related to a quadratic parametric optimization problem involving the second order derivatives at the solution of the original problem. We perform a general abstract analysis of inexact SGD recursions, that is, SGD with an additional error term which is not required to have zero mean. This constitutes a result of independent interest, which we apply to the sequence of SGD derivatives in order to prove their convergence toward the derivative of the solution mapping. The developed theory is illustrated with numerical experiments on synthetic tasks. We believe our work paves the way to a better understanding of stochastic hyperparameter optimization, and more generally stochastic meta-learning strategies.

**Related works.** Differentiating through algorithms is closely associated with the broader concept of *automatic differentiation* (Griewank, 1989). In practice, it is implemented using either the forward mode (Wengert, 1964), or the more common reverse mode (Rumelhart et al., 1986) known as backpropagation. For detailed surveys, see (Griewank et al., 1993) or (Griewank and Walther, 2008; Baydin et al., 2018). Modern machine learning is intrinsically linked to this idea through the use of Python frameworks like Tensorflow (Abadi et al., 2015), PyTorch (Paszke et al., 2019), and JAX (Bradbury et al., 2018; Blondel et al., 2022). When using the reverse mode, a limitation of this method is the need to retain every iteration of the inner optimization process in memory, although this challenge can be mitigated by employing checkpointing, invertible optimization algorithms (Maclaurin et al., 2015), by utilizing truncated backpropagation (Shaban et al., 2019), Jacobian-free backpropagation (Fung et al., 2022) or one-step differentiation (Bolte et al., 2023).

Along with iterative differentiation (ITD), (approximate) implicit differentiation (AID) plays an increasing important role, sometimes under the name implicit deep learning. El Ghaoui et al. (2021) highlights the utility of fixed-point equations in defining hidden features, and (Bai et al., 2019) proposes equilibrium points for sequence models, reducing memory consumption significantly. Further, (Bertrand et al., 2020; Agrawal et al., 2019) expands implicit differentiation's applications to high-dimensional, non-smooth problems and convex programs. Ablin et al. (2020) emphasizes the computational benefits of automatic differentiation, particularly in min-min optimization. In particular, OptNet (Amos and Kolter, 2017) and Deep Equilibrium Models (DEQ) (Bai et al., 2019) are examples of relevant applications.

Hypergradient estimation through iterative differentiation or implicit differentiation has a long story in machine learning (Pedregosa, 2016; Lorraine et al., 2020). In the context of imaging, itera-

tive differentiation was used to perform hyperparameter selection through the Stein's unbiased risk estimator (Deledalle et al., 2014), and also for refitting procedure (Deledalle et al., 2017). Model-agnostic Meta-learning (MAML) was introduced by Finn et al. (2017) as a methodology to train neural architectures that adapt to new tasks through iterative differentiation (meta-learning). It was later adapted to implicit differentiation (Rajeswaran et al., 2019). These developments motivated further studies of the bilevel programming problem in a machine learning context (Franceschi et al., 2018; Grazzi et al., 2020).

The literature on the stochastic iterative and implicit differentiation is more limited. In the stochastic setting, Grazzi et al. (2021, 2023, 2024) considered implicit differentiation, mostly as a stochastic approximation to solve the implicit differentiation linear equation or use independent copies for the derivative part. In general stochastic approaches for bilevel optimization sample different batches for the iterate and derivative recursions. Here we *jointly analyze both recursion* with the same samples. Despite this lack of systemic theoretical analysis of convergence, differentiating through the SGD iterates is mentioned in Maclaurin et al. (2015) which is focused on an efficient implementation of backpropagation through SGD, Pedregosa (2016) which explicitly calls for the development of differentiation techniques for stochastic optimization algorithms. Furhtermore Finn et al. (2017); Ji et al. (2021) suggests explicitly to use differentiation through stochastic first order solvers and this was further explicitly considered by Ji et al. (2022) in a meta-learning context.

Closely related to the general issue of differentiating parametric optimization problems is solving bilevel optimization, where the Jacobian of the inner problem is crucial to analyze. Chen et al. (2021) introduces a method, demonstrating improved convergence rates for stochastic nested problems through a unified SGD approach. In the same vein, Arbel and Mairal (2021) leverages inexact implicit differentiation and warm-start strategies to match the computational efficiency of oracle methods, proving effective in hyperparameter optimization. Additionally, the work (Ji et al., 2021) provides a thorough convergence analysis for AID and ITD-based methods, proposing the novel stocBiO algorithm for enhanced sample complexity. Furthermore, (Dagréou et al., 2022; Dagréou et al., 2024) introduce a novel framework allowing unbiased gradient estimates and variance reduction methods for stochastic bilevel optimization.

Although this is not the initial focus of this work, the technical bulk of our arguments requires an analysis of *perturbed, or inexact, SGD sequences*. This amounts to study the robustness of the stochastic gradient algorithm with non-centered noise, or equivalently non-vanishing deterministic errors. Such questioning around robustness to errors have existed for decades in the stochastic approximation literature, see for example (Ermoliev, 1983; Chen et al., 1987) and references therein. Many existing results presented in the literature are qualitative and relate to nonconvex optimization (Solodov and Zavriev, 1998; Borkar, 2009; Doucet and Tadic, 2017; Ramaswamy and Bhatnagar, 2017; Dieuleveut et al., 2023). Let us also mention the smooth convex setting for which inexact oracles have been studied by (Nedić and Bertsekas, 2010; Devolder et al., 2014). A recent account of existing convergence results for biased SGD is given by Demidovich et al. (2023). As a by-product of our arguments, we provide a general mean squared error convergence analysis of inexact SGD for a diversity of step size regimes, in the smooth, strongly convex setting. Our analysis allows to handle random non stationary bias terms, whose magnitude depend on the iteration counter $k$. This is customary as the errors in the sequence of derivatives are due to the suboptimality of the sequence of iterates. These errors thus depend on the realization of the iterate sequence, requiring a dedicated analysis not covered by existing art Demidovich et al. (2023).

## 2 The derivative of SGD is inexact SGD

### 2.1 Intuitive overview

We consider a parametric stochastic optimization problem of the form

$$x^\star(\theta) = \arg\min_{x \in \mathbb{R}^d} \ F(x, \theta) \coloneqq \mathbb{E}_{\xi \sim P}[f(x, \theta; \xi)] \tag{Opt}$$

where $F \colon \mathbb{R}^d \times \Theta \to \mathbb{R}$ is smooth and strongly convex in $x$ for a fixed $\theta$. The stochastic gradient descent algorithm, stochastic gradient descent (SGD), is defined by an initialization $x_0(\theta)$, and for $k \in \mathbb{N}$

$$x_{k+1}(\theta) = x_k(\theta) - \eta_k \nabla_x f(x_k(\theta), \theta; \xi_{k+1}) \tag{SGD}$$

where $(\eta_k)_{k\in\mathbb{N}}$ is a sequence of positive step-sizes and $(\xi_k)_{k\in\mathbb{N}}$ is a sequence of independent random variables with common distribution P. Precise assumptions on the problem and the algorithm will be given in Section 2.2 to ensure convergence. We highlight here that both the objective $f(x,\theta,\xi)$ and the initialization of the algorithm $x_0(\theta)$ depend on some parameter $\theta \in \Theta \subset \mathbb{R}^m$, and so do the iterates and optimal solution.

For any $\theta \in \Theta$ and any $k \geq 0$, under appropriate assumptions, the Jacobian of $x_k(\theta)$ w.r.t. $\theta$, denoted by $\partial_\theta x_k(\theta) \in \mathbb{R}^{d\times m}$, is well defined and obeys the following recursion from the chain rule of differentiation:

$$\partial_\theta x_{k+1}(\theta) = \partial_\theta x_k(\theta) - \eta_k \nabla^2_{xx} f(x_k(\theta), \theta; \xi_{k+1}) \partial_\theta x_k(\theta) - \eta_k \nabla^2_{x\theta} f(x_k(\theta), \theta; \xi_{k+1}). \quad \text{(SGD')}$$

The natural limit candidate for this recursion is the Jacobian of the solution, $\partial_\theta x^\star(\theta)$, which, from the implicit function theorem, is the unique solution to the following linear system

$$\nabla^2_{xx} F(x^\star(\theta), \theta) D + \nabla^2_{x\theta} F(x^\star(\theta), \theta) = \mathbb{E}_{\xi\sim P}\big[\nabla^2_{xx} f(x^\star(\theta), \theta; \xi) D + \nabla^2_{x\theta} f(x^\star(\theta), \theta; \xi)\big] = 0.$$

As noted in (Arbel and Mairal, 2021, Proposition 1), this is equivalently characterized as a solution to the following stochastic minimization problem

$$\partial_\theta x^\star(\theta) = \arg\min_{D\in\mathbb{R}^{d\times m}} \mathbb{E}_{\xi\sim P}\left[\left\langle \frac{1}{2}\nabla^2_{xx} f(x^\star(\theta), \theta; \xi) D + \nabla^2_{x\theta} f(x^\star(\theta), \theta; \xi), D \right\rangle\right] \quad \text{(Opt')}$$

where we use the standard Frobenius inner product over matrices. Our key insight is to formally understand the recursion in (SGD') as an inexact SGD sequence applied to problem (Opt').

**Intuition from the quadratic case.** Consider two maps $\xi \mapsto Q(\xi) \in \mathbb{R}^{d\times d}$ and $\xi \mapsto B(\xi) \in \mathbb{R}^{d\times m}$. Let $f(x, \theta; \xi) = \frac{1}{2}x^\top Q(\xi)x + x^\top B(\xi)\theta$, then the recursion in (SGD') becomes

$$\partial_\theta x_{k+1}(\theta) = \partial_\theta x_k(\theta) - \eta_k (Q(\xi_{k+1})\partial_\theta x_k(\theta) + B(\xi_{k+1})).$$

which is exactly a stochastic gradient descent sequence for problem (Opt'). Hence, choosing appropriate step sizes ensures convergence. Beyond the quadratic setting, one needs to take into consideration the fact that the second order derivatives of $f$ are not constant, leading to our interpretation as *perturbed* stochastic gradient iterates for the derivatives, as detailed below.

**The general case.** We rewrite the recursion (SGD') as follows

$$\partial_\theta x_{k+1}(\theta) = \partial_\theta x_k(\theta) - \eta_k \nabla^2_{xx} f(x^\star(\theta), \theta; \xi_{k+1}) \partial_\theta x_k(\theta) - \eta_k \nabla^2_{x\theta} f(x^\star(\theta), \theta; \xi_{k+1}) + e_{k+1}, \quad (1)$$

where the error term $e_{k+1}$ is defined as

$$\begin{aligned}
e_{k+1} = &\; \eta_k \left(\nabla^2_{xx} f(x^\star(\theta), \theta; \xi_{k+1}) - \nabla^2_{xx} f(x_k(\theta), \theta; \xi_{k+1})\right) \partial_\theta x_k(\theta) \\
&+ \eta_k \left(\nabla^2_{x\theta} f(x^\star(\theta), \theta; \xi_{k+1}) - \nabla^2_{x\theta} f(x_k(\theta), \theta; \xi_{k+1})\right).
\end{aligned}$$

Assuming that the second derivative of $f$ is Lipschitz-continuous, the error term $e_{k+1}$ is of the same order as $\eta_k \|x_k(\theta) - x^\star(\theta)\|(1 + \|\partial_\theta x_k(\theta)\|)$. Our main contribution is a careful analysis of a specific version of inexact SGD which covers the above recursion. Under typical stochastic approximation assumptions, the convergence of $x_k(\theta)$ toward $x^\star(\theta)$ essentially entails the convergence of $\partial_\theta x_k(\theta)$ toward $\partial_\theta x^\star(\theta)$. This allows us to carry out a joint convergence analysis of both sequences in (SGD) and (SGD'). We now describe the assumptions required to make this intuition rigorous.

## 2.2 Main assumptions

We start with the stochastic objective, $f$ in (Opt) and then specify assumptions on the underlying random variable $\xi$. The crucial assumption for our results is strong convexity. The rest of the assumptions are typically satisfied in applications such as hyper parameter tuning. We point out that both examples in the numerical section satisfy our assumptions and are implemented in the regime described by our main theorem.

**Assumption 1.** Let $\Theta$ be an open Euclidean subset of $\mathbb{R}^m$ and $\Xi$ be a measure space. The function $f: \mathbb{R}^d \times \Theta \times \Xi \to \mathbb{R}$ satisfies the following conditions:

(a) *Differentiability:* $f(\cdot, \cdot; \xi)$ is $C^2$, with $M$-Lipschitz Hessian (in Frobenius norm), for all $\xi \in \Xi$.

(b) *Smoothness:* $\nabla_x f(\cdot, \theta; \xi)$ is $L$-Lipschitz and $\nabla_x f(x, \cdot; \xi)$ is $L'$-Lipschitz for all $x, \theta$ and $\xi \in \Xi$.

(c) *Strong convexity:* $f(\cdot, \theta; \xi)$ is $\mu$-strongly convex for all $\theta \in \Theta$ and $\xi \in \Xi$.

Assumption 1(b) entails that $\nabla^2_{xx} f$ and $\nabla^2_{x\theta} f$ are uniformly bounded in operator norm by $L$ and $L'$ respectively. We remark that our smoothness assumption is global in $x$, but possibly only local in $\theta$ since $\Theta$ is an arbitrary open neighborhood, so that Assumption 1(b) does not require global Lipschicity with respect to the variable $\theta$. Assumption 1(c) implies that $F(\cdot, \theta)$ has a unique minimizer that we will denote by $x^\star(\theta)$; it also implies that $\nabla^2_{xx} f$ is positive definite. This is actually the strongest part of Assumption 1, it is somewhat a requirement since the problem of differentiating the solution to an optimization problem, not necessarily strongly convex, is not settled for the moment.

As a consequence of Assumption 1, the derivative sequence in (SGD') is almost surely bounded[1]. This is proved in Appendix B.

**Lemma 2.1.** *Under Assumption 1, assuming that $\eta_k \leq \frac{1}{L}$ for all $k$, we have almost surely* $\|\partial_\theta x_k(\theta)\| \leq \max\{\|\partial_\theta x_0(\theta)\|, \sqrt{m}L'/\mu\}$.

We now specify the structure of the random variables $(\xi_k)_{k \in \mathbb{N}}$ appearing in the recursions (SGD) and (SGD'). In particular, we follow the classical approach of (Bottou et al., 2018; Gower et al., 2019) among a rich literature for our variance condition.

**Assumption 2.** The observed noise sequence $(\xi_k)_{k \in \mathbb{N}}$ is independent identically distributed with common distribution P on $\Xi$. Furthermore,

(a) *Variance control:* there is $\sigma \geq 0$ such that for all $\theta \in \Theta$,
$$\mathbb{E}\big[\|\nabla_x f(x^\star(\theta), \theta; \xi)\|^2\big] \leq \sigma^2, \quad \mathbb{E}\big[\|\nabla^2_{xx} f(x^\star(\theta), \theta; \xi)\partial_\theta x^\star(\theta) + \nabla^2_{x\theta} f(x^\star(\theta), \theta; \xi)\|^2\big] \leq \sigma^2.$$

(b) *Integrability:* $f(x, \theta; \cdot)$ and $\nabla f(x, \theta; \cdot)$ are integrable w.r.t. P for a certain fixed pair $x \in \mathbb{R}^d$, $\theta \in \Theta$.

Note that we control the second moment only *at the solution*, which means that the case $\sigma^2 = 0$ corresponds to the interpolation scenario but does *not* mean that the algorithm is noiseless. Furthermore, we also control the second moment of the second derivative (in Frobenius norm). This is not typical in the SGD literature but is required here to analyze the sequence of derivatives (this is illustrated in the *simple interpolation* case of Fig. 1). Assumption 1(a) and (b) together with Assumption 2 imply that one can permute expectation and derivative up to order 2, as detailed in Appendix A.

In this setting, we use the natural filtration $(\mathcal{F}_k)_{k \in \mathbb{N}}$ where for all $k$, $\mathcal{F}_k$ is defined as the $\sigma$-algebra generated by $\xi_0, \ldots, \xi_k$. Note that $\xi_{k+1}$ and thus $\nabla_x f(x_k(\theta), \theta; \xi_{k+1})$ is not $\mathcal{F}_k$-measurable but $\mathcal{F}_{k+1}$-measurable.

## 2.3 Main result on the convergence of the derivatives of SGD

The following is the main result of this paper. Its proof is postponed to Section 3.2.

**Theorem 2.2** (Convergence of the derivatives of SGD). *Let $\Theta \subset \mathbb{R}^m$ be open, $\Xi$ be a measure space and $f : \mathbb{R}^d \times \Theta \times \Xi \to \mathbb{R}$ be as in Assumption 1. Set $\kappa = L/\mu$, the condition number. Let $(\xi_k)_{k \in \mathbb{N}}$ be a sequence of independent variables on $\Xi$, as in Assumption 2. Let $(\eta_k)_{k \in \mathbb{N}}$ be a positive, non-increasing, non-summable sequence with $\eta_0 \leq \frac{\mu}{4L^2} = \frac{1}{\mu}\frac{1}{4\kappa^2}$ and $(x_k(\theta))_{k \in \mathbb{N}}$ be defined as in (SGD). Then:*

• *General estimates: setting $\eta = \lim_{k \to \infty} \eta_k$, we have*
$$\limsup_{k \to \infty} \quad \mathbb{E}\big[\|\partial_\theta x_k(\theta) - \partial_\theta x^\star(\theta)\|^2\big] \leq \frac{4\sigma^2\eta}{\mu}\left(1 + \frac{3M(1 + \max\{\|\partial_\theta x_0(\theta)\|, \sqrt{m}L'/\mu\})}{\mu}\right)^2.$$

• *Sublinear rate: if for all $k$, $\eta_k = \frac{1}{\mu}\frac{2}{k+8\kappa^2}$, then*
$$\mathbb{E}\big[\|\partial_\theta x_k(\theta) - \partial_\theta x^\star(\theta)\|^2\big] = O\left(\frac{\log(k + 8\kappa^2)^2}{k + 8\kappa^2}\right).$$

---
[1] This does not depend on the randomness structure detailed in Assumption 2.

*where the constants in the big O are polynomials in $\kappa$, $\|x_0(\theta) - x^\star(\theta)\|^2$, $\|\partial_\theta x_0(\theta) - \partial_\theta x^\star(\theta)\|^2$,*
*$\sigma^2$, $\frac{1}{\mu}$, $M$ and $\sqrt{m}$.*

• *Interpolation regime: if $\sigma = 0$ and $\eta_k = \frac{\mu}{4L^2}$ for all $k \in \mathbb{N}$, then*

$$\mathbb{E}\big[\|\partial_\theta x_k(\theta) - \partial_\theta x^\star(\theta)\|^2\big] = O\left(k\left(1 - \frac{1}{8\kappa^2}\right)^k\right).$$

The first part of the result provides a general estimate which allows covering virtually all small step-size cases. This includes: i) vanishing step-sizes, for which our result implies convergence of derivatives; and ii) constant step-sizes $\eta$, for which we provide a bound on the distance to the true derivative that is proportional to $\eta$. For the second part, using step-sizes decreasing as $1/k$, which is a typical setup for the convergence of SGD on strongly convex objectives, our result shows that the derivatives converge as well, with a rate that is asymptotically of the same order, up to log factors. Finally, the last part of the result relates to the interpolation regime which has drawn a lot of attention in recent years because it captures some of the features of overparameterized deep neural network training Ma et al. (2018); Varre et al. (2021); Garrigos and Gower (2023). Note that the condition $\sigma = 0$ in Assumption 2 entails that interpolation occurs for both problems (Opt) and (Opt'), and in this case we obtain exponential convergence of both the iterates and their derivatives, with a constant stepsize, as in the deterministic setting (Mehmood and Ochs, 2020).

**Remark 2.3.** *The specific stepsize used to obtain the sublinear rate actually applies to any stepsize of the form $\eta_k = 2/(ck + 8u)$ for given $c, u > 0$ such that $0 < c \leq \mu$ and $u \geq L^2/c$. One obtains the same result with $\mu, L$ respectively replaced in the expressions by $\tilde{\mu} := c \leq \mu$ and $\tilde{L} := \sqrt{uc} \geq L$. This corresponds to using a lower estimate for the strong convexity constant and a higher estimate for the smoothness constant, which remain valid. A similar remark holds for the interpolation regime where any stepsize $\eta$ smaller than $\mu/(4L^2)$ will bring the same result with $\kappa$ replaced by $\tilde{\kappa} := 1/(4L\eta)$ in the statement.*

**Remark 2.4.** *We consider step sizes at most $\frac{\mu}{4L^2}$ which is smaller than $\frac{1}{L}$, typically used in optimization. Aside from the $\frac{1}{4}$ factor, which could possibly be improved, it is important to relate it to the the fact that we have obtain $O(1/k)$ rates which represent fast rates for SGD for convex optimization, limited to strongly convex objectives. Second, we do not have any Lipschicity assumption on the objective function itself. This, and the use of small steps to obtain fast rate is in line with related literature such as (Bottou et al., 2018, Theorem 4.6), the discussion following (Moulines and Bach, 2011, Theorem 1) or (Garrigos and Gower, 2023, Corollary 5.8 and Theorem 5.9). The possibility to obtain convergence of derivatives of SGD for larger step sizes will be a topic of future research.*

## 3 Proof of the main result

Our result relies on the interpretation of the recursion (SGD') as an inexact SGD sequence for the problem (Opt'). We start with a detailed analysis of inexact SGD under appropriate assumptions. This is an abstract result which we formulate using an abstract function $g$ different from the objective in problems (Opt) and (Opt') in order to avoid any possible confusion. In particular $g$ is static (does not depend on external parameters) and the obtained convergence result will be then applied to both sequences (SGD) and (SGD').

### 3.1 Detour through an auxiliary result: convergence of inexact SGD

We provide here our template results for the convergence of inexact SGD. As template, we consider a function $G: \mathbb{R}^q \to \mathbb{R}$ defined as

$$G(x) := \mathbb{E}_{\xi \sim P}[g(x; \xi)].$$

Our generic assumptions stand as follows.

**Assumption 3.** P *is a probability distribution on the measure space $\Xi$, and the function $g: \mathbb{R}^d \times \Xi \to \mathbb{R}$ satisfies the following conditions:*

(a) *Smoothness: $g(\cdot; \xi)$ is $C^1$ with $L$-Lipschitz gradient, i.e., there is $L \geq 0$ such that*
$$\|\nabla_x g(x; \xi) - \nabla_x g(x'; \xi)\| \leq L\|x - x'\|$$
*for all $x, x' \in \mathbb{R}^q$, and all $\xi \in \Xi$.*

(b) *Strong convexity:* there is $x^\star \in \mathbb{R}^q$ and $\mu > 0$ such that $\langle x - x^\star, \mathbb{E}[\nabla_x g(x; \xi)]\rangle \geq \mu \|x - x^\star\|^2$ for all $x \in \mathbb{R}^q$.

(c) *Variance control:* there is $0 \leq \sigma < +\infty$ such that $\mathbb{E}\big[\|\nabla_x g(x^\star; \xi)\|^2\big] \leq \sigma^2$.

We remark that under Assumptions 1 and 2, Assumption 3 is satisfied for both problems (Opt) and (Opt'). We will consider an *inexact* SGD recursion of the form

$$x_{k+1} = x_k - \eta_k \left(\nabla_x g(x_k; \xi_{k+1}) + e_{k+1}\right) \tag{2}$$

where we will need the following assumption on noise and errors.

**Assumption 4.** The observed noise sequence $(\xi_k)_{k \in \mathbb{N}}$ is independent and identically distributed with common distribution P on $\Xi$. Denote by $(\mathcal{F}_k)_{k \in \mathbb{N}}$ the natural filtration (i.e., for all $k$, $\mathcal{F}_k$ is the $\sigma$-algebra generated by $\xi_0, \ldots, \xi_k$), the errors $(e_k)_{k \in \mathbb{N}}$ form a sequence of $(\mathcal{F}_k)_{k \in \mathbb{N}}$-adapted random variables such that $\mathbb{E}[\|e_{k+1}\|^2] \leq B_k^2$ where $(B_k)_{k \in \mathbb{N}}$ is a deterministic non-increasing sequence.

The following reduces the analysis of inexact SGD sequences to the study of a deterministic recursion, its proof is given in Appendix B.

**Proposition 3.1** (Convergence of inexact SGD). *Let Assumption 3 and Assumption 4 hold. Consider the iterates in* (2) *where* $(\eta_k)_{k \in \mathbb{N}}$ *is a positive, non-increasing, non-summable sequence with* $\eta_0 \leq \frac{\mu}{4L^2}$. *Setting* $D_k = \sqrt{\mathbb{E}[\|x_k - x^\star\|^2]}$, *we have for all* $k \in \mathbb{N}$:

$$D_{k+1}^2 \leq (1 - \mu \eta_k) D_k^2 + 2\eta_k^2 (B_k^2 + 2\sigma^2) + 2\eta_k B_k D_k. \tag{3}$$

Studying the deterministic recursion (3) leads to the following results by relying on different helper lemmas laid out in Appendix C:

| Lemma | Stepsizes | Errors | Noise | Result |
|-------|-----------|--------|-------|--------|
| Lemma C.1 | $\eta_k \to \eta \geq 0$ | $B_k \to B \propto \sqrt{\eta}$ | $\sigma^2 \geq 0$ | $\limsup_{k \to \infty} \quad D_k \propto \sqrt{\eta}$ |
| Lemma C.2 | $\eta_k = \frac{2\mu}{\mu^2 k + 8L^2}$ | $B_k = 0$ | $\sigma^2 \geq 0$ | $D_k^2 = O\left(\frac{\log(k + 8\kappa^2)}{k + 8\kappa^2}\right)$ |
| Lemma C.3 | $\eta_k = \frac{2\mu}{\mu^2 k + 8L^2}$ | $B_k^2 = O\left(\frac{\log(k + 8\kappa^2)}{k + 8\kappa^2}\right)$ | $\sigma^2 \geq 0$ | $D_k^2 = O\left(\frac{\log(k + 8\kappa^2)^2}{k + 8\kappa^2}\right)$ |
| Lemma C.4 | $\eta_k = \eta < \frac{1}{2\mu}$ | $B_k^2 = O\left((1 - \frac{\mu\eta}{2})^k\right)$ | $\sigma^2 = 0$ | $D_k^2 = O\left(k(1 - \frac{\mu\eta}{2})^k\right)$ |

These results will be used to prove Theorem 2.2 in the coming section. They are of independent interest regarding the convergence analysis of inexact SGD sequences. The first lemma allows to prove the first point in Theorem 2.2, the second and third lemmas allow to treat the second point, and the last lemma allows to treat the interpolation regime in the third point. See Appendix C for detailed statements.

## 3.2 Proof of the main result

We first show that Proposition 3.1 can be applied to the recursion (SGD') in relation to (Opt') and then explicit its consequences using the lemmas of Appendix C.

*Proof of Theorem 2.2.* Following (1), we have that $(\partial_\theta x_k(\theta))_{k \in \mathbb{N}}$ is an inexact SGD sequence for problem (Opt') as in (2), with an error term of the form

$$e_{k+1} = \left(\nabla_{xx}^2 f(x^\star(\theta), \theta; \xi_{k+1}) - \nabla_{xx}^2 f(x_k(\theta), \theta; \xi_{k+1})\right) \partial_\theta x_k(\theta)$$
$$+ \left(\nabla_{x\theta}^2 f(x^\star(\theta), \theta; \xi_{k+1}) - \nabla_{x\theta}^2 f(x_k(\theta), \theta; \xi_{k+1})\right).$$

Under Assumption 1 and Assumption 2, Problem (Opt') satisfies Assumption 3, and we have the same values for $L$, $\mu$ and $\sigma$ for both problems (Opt) and (Opt'). Furthermore, the error term $e_{k+1}$ satisfies Assumption 4, and, thanks to Lemma 2.1 and Assumption 1 on Lipschitz continuity of the Hessian of $f$, we have almost surely

$$\|e_{k+1}\| \leq M \|x_k(\theta) - x^\star(\theta)\|(1 + \max\{\|\partial_\theta x_0(\theta)\|, \sqrt{m} L'/\mu\}). \tag{4}$$

The various bounds are obtained by considering different regimes. We first estimate a bound on $\mathbb{E}\big[\|x_k(\theta) - x^\star(\theta)\|^2\big]$ using Proposition 3.1 with $B_k = 0$ for all $k$. This allows to obtain an estimate

on $\mathbb{E}\big[\|e_{k+1}\|^2\big]$ using (4). We conclude for the derivative sequence by applying Proposition 3.1 with its different corollaries. We treat all these results separately.

**General estimate.** From Proposition 3.1 with $B_k = 0$, we obtain, by considering $g(x, \xi) = f(x, \theta; \xi)$ and Lemma C.1 that $\limsup_{k\to\infty} \mathbb{E}\big[\|x_k(\theta) - x^\star(\theta)\|^2\big] \leq \frac{4\sigma^2\eta}{\mu}$. For the derivative sequence, combining this first estimate with (4), we can consider a decreasing sequence of mean squared upper bounds $(B_k)_{k\in\mathbb{N}}$, such that

$$\lim_{k\to\infty} B_k = B := 2\sigma\sqrt{\frac{\eta}{\mu}}M(1 + \max\{\|\partial_\theta x_0(\theta)\|, \sqrt{m}L'/\mu\}).$$

The upper bound given by Proposition 3.1 and Lemma C.1 is of the form

$$\frac{\sqrt{B^2 + 2\mu\eta(B^2 + 2\sigma^2)} + B}{\mu} \leq \frac{\sqrt{\frac{3}{2}B^2 + 4\mu\eta\sigma^2} + B}{\mu} \leq 2\sigma\sqrt{\frac{\eta}{\mu}} + \frac{3B}{\mu},$$

which corresponds to the claimed bound.

**Convergence rate.** From Proposition 3.1 with $B_k = 0$, we obtain, by considering $g(x, \xi) = f(x, \theta; \xi)$ and Lemma C.2 that $\mathbb{E}\big[\|x_k(\theta) - x^\star(\theta)\|^2\big] = O\left(\frac{\log(k+8\kappa^2)}{k+8\kappa^2}\right)$ as given in Lemma C.2. As a consequence, combining this first estimate with (4), we may set $B_k = O\left(\frac{\log(k+8\kappa^2)}{k+8\kappa^2}\right)$ and the result follows from Lemma C.3.

**Interpolation regime.** Setting $\rho = 1 - \frac{\mu\eta}{2} = 1 - \frac{1}{8\kappa^2}$, for $\sigma^2 = 0$ and $B_k = 0$ for all $k \in \mathbb{N}$, it is clear from (3) that $\mathbb{E}\big[\|x_k(\theta) - x^\star(\theta)\|^2\big] \leq \|x_0(\theta) - x^\star(\theta)\|^2\rho^k$ for all $k \in \mathbb{N}$. Using (4), we may choose $B_k = O(\rho^k)$. Plugging this estimate in (3), the result is then given by Lemma C.4. $\square$

## 4    Numerical illustration

In this section, we illustrate the results of Theorem 2.2 by examining the numerical behavior of the iterates and their derivatives under various settings. Specifically, we provide insights into the behavior of classical regularized methods, such as Ridge regression, logistic regression, Huber regression. Furthermore, we explore potential extensions to the nonsmooth case by also considering the Hinge loss. All the experiments are performed for the empirical risk minimization structure, i.e., the randomness $\xi$ is drawn from the uniform distribution over $\{1, \ldots, m\}$. All the experiments were performed in `jax` (Bradbury et al., 2018) on a MacBook Pro M3 Max.

**Ordinary least squares.** We consider a simple linear regression problem solved by ordinary least-squares as:

$$x^\star(\theta) = \arg\min_{x\in\mathbb{R}^d} F(x, \theta) := \frac{1}{2m}\sum_{\xi=1}^m (a_\xi^\top x - b(\theta)_\xi)^2,$$

The data $A = (a_\xi) \in \mathbb{R}^{m\times d}$ here is a random matrix with $d < m$. The finite sum structure naturally suggests a stochastic gradient decomposition as in (SGD), by choosing $\xi$ uniformly in $\{1, \ldots, m\}$ with replacement. We consider three generative models for the function $b$:

1. *Standard setting:* $\theta \in \mathbb{R}^m$, and we have $b$ is the identity on $\mathbb{R}^m$. In this case, our theory corresponds to the differentiation of the least squares solution seen as a function of the output observations.

2. *Simple interpolation setting:* The setting is the same as the standard one, except that we consider a specific value of $\theta = A\zeta$ for some $\zeta \in \mathbb{R}^d$. In this case, we do *not* differentiate through the linear relation $\theta = A\zeta$, the function $b$ remains the identity, we simply evaluate at a specific point which corresponds to data interpolation. We call this simple interpolation, because it corresponds to $\sigma = 0$ for the sequence (SGD), but not for (SGD').

3. *Double interpolation setting:* The parameter variable $\theta$ is in $\mathbb{R}^d$ and we set $b: \theta \to A\theta$. Here this corresponds to an interpolation regime which is uniform in $\theta$. We call this double interpolation because it corresponds to $\sigma = 0$ for both sequences (SGD) and (SGD').

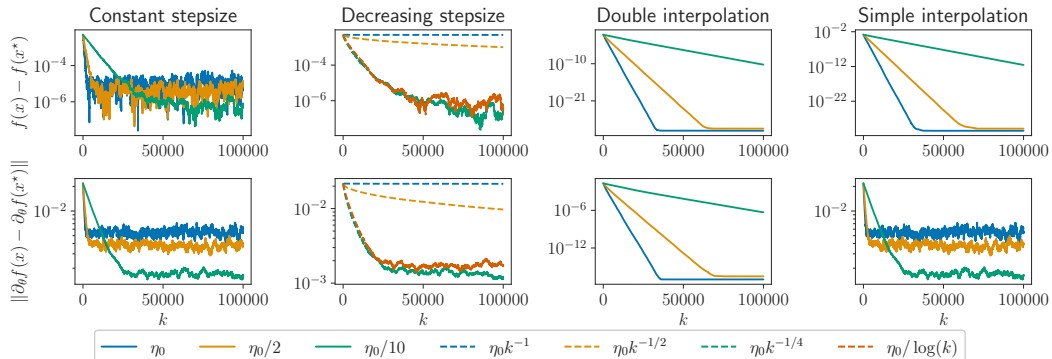

**Figure 1:** Numerical behavior of SGD iterates and their derivatives (Jacobians) in a linear regression problem solved by ordinary least squares. The plots depict the convergence of the suboptimality $f(x_k(\theta)) - f(x^\star(\theta))$ and the Frobenius norm of the derivative error $\|\partial_\theta x_k(\theta) - \partial_\theta x^\star(\theta)\|_F$ across different experimental settings: constant step-size (first column), decreasing step-size (second column), double interpolation (third column), and simple interpolation (fourth column). The experiments utilize varying step-size strategies to illustrate general estimates, sublinear rates, and the impacts of interpolation regimes, validating theoretical predictions of Theorem 2.2.

Note the the difference between setting 2. and 3. are that we are *not* differentiating through the linear map $A$ in setting 2. Furthermore Assumption 1 and Assumption 2 are satisfied for these three settings. Figure 1 illustrates the behavior of (SGD) and (SGD'). More precisely, we monitor the convergence of the suboptimality $f(x_k(\theta)) - f(x^\star(\theta))$ and of the derivatives error measured in Frobenius norm $\|\partial_\theta x_k(\theta) - \partial_\theta x^\star(\theta)\|_F$. We consider various step size regimes and set $\eta_0 = \frac{\mu}{4L^2}$ for all experiments. This allow us to clearly identify the three regimes of Theorem 2.2:

- *Constant stepsize:* in setting 1., employing a constant step-size, we observe convergence of both the iterates (consistent with classical SGD theory) and their derivatives to a neighborhood of the solution whose diameter decreases with the step size.

- *Decreasing stepsize:* in setting 1., employing a step-size proportional to $1/k$, we observe a sublinear decay of both the iterates and their derivatives. The convergence is difficult to observe since the decay leads to very small updates.

- *Double Interpolation regime:* in setting 3., employing a constant step-size, we observe both iterates and derivatives linear decays.

- *Simple Interpolation regime:* in setting 2., Assumption 2(a) is satisfied with $\sigma = 0$ only for the iterates, but not for the derivatives, we observe linear convergence of the iterates, but the derivatives converge to a neighborhood of the solution as in the setting 1.

**Ridge, Logistic, Huber and SVM regression.** In addition to the previous illustration of Theorem 2.2, we provide numerical experiments for constant learning rate for four different models: ridge regression, logistic regression, Huber regression and Support Vector Machines (SVM) regression. All of them are written as

$$x^\star(\theta) = \arg\min_{x \in \mathbb{R}^d} \ F(x, \theta) := \frac{1}{m} \sum_{\xi=1}^{m} f(x, \theta; \xi) + \mu\|x\|_2^2,$$

where $f(x, \theta; \xi) = \frac{1}{2}(a_\xi^\top w - \theta_\xi)^2$ for ridge regression, $f(x, \theta; \xi) = \log(1 + \exp(-\theta_\xi a_\xi^\top x))$ for logistic regression,

$$f(x, \theta; \xi) = \begin{cases} \frac{1}{2}(\theta_\xi - a_\xi^\top x)^2 & \text{if } |\theta_\xi - a_\xi^\top x| \le \delta \\ \delta\left(|\theta_\xi - a_\xi^\top x| - \frac{1}{2}\delta\right) & \text{otherwise,} \end{cases}$$

for Huber regression for some $\delta > 0$ (here $\delta = 0.1$), and $f(x, \theta; \xi) = \max(0, 1 - \theta_\xi a_\xi^\top x)$ for SVM regression (hinge loss). In all cases, the finite sum structure naturally suggests a stochastic gradient decomposition as in (SGD), by choosing $\xi$ uniformly in $\{1, \ldots, m\}$ with replacement All experiences are performed with $m, d = 100, 10$ and $\mu = 0.05$. In Figure 2, we show the convergence

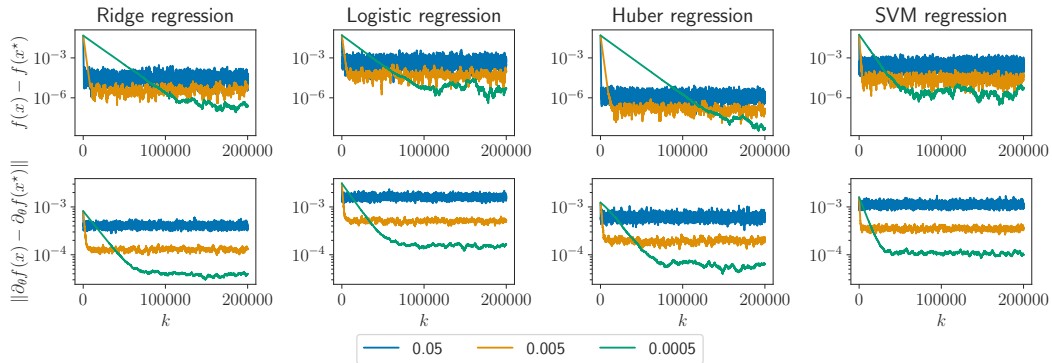

**Figure 2:** Numerical behavior of the objective function and its derivatives with respect to $\theta$ for ridge regression, logistic regression, Huber regression, and Support Vector Machines (SVM) regression using a constant learning rate. We report the suboptimality $f(x_k(\theta)) - f(x^\star(\theta))$ for the SGD iterates, along *(bottom)* with the norm of derivatives errors $\|\partial_\theta x_k(\theta) - \partial_\theta x^\star(\theta)\|_F$ for different constant step-size. Each line corresponds to a different step-size.

of the objective function and the derivatives with respect to $\theta$ for the four models with a constant learning rate. Note that the SVM loss is not differentiable. We refer to (Bolte et al., 2022) for a formal treatment of nonsmooth iterative differentiation, but one could expect similar results for conservative Jacobians.

**Experiments on real data.** We display in Figure 3 the behaviour of SGD iterates and their derivatives for regularized logistic regression problem on `ijcnn1`.

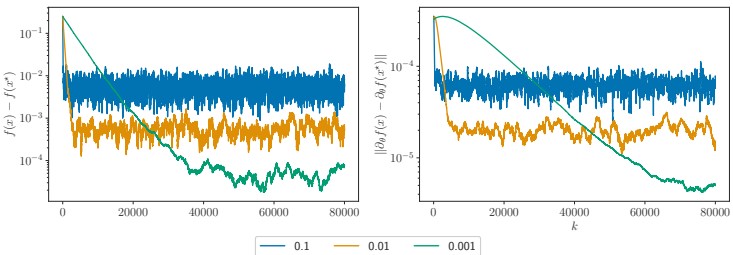

**Figure 3:** Numerical behavior of SGD iterates and their derivatives (Jacobians) for regularized logistic regression problem. The plots depict the convergence of the suboptimality $f(x_k(\theta)) - f(x^\star(\theta))$ (left) and the derivative error $\|\partial_\theta x_k(\theta) - \partial_\theta x^\star(\theta)\|$ (right) for different constant step size. The dataset used is `ijcnn1` from `libsvm` with 49,990 observations and 22 features. The observations are qualitatively identical to our synthetic experiments.

## 5   Conclusion

In conclusion, our study of stochastic optimization problems where the objective depends on a parameter reveals insights into the behavior of SGD derivatives. We demonstrated that these derivatives follow an inexact SGD recursion, converging to the solution mapping's derivative under strong convexity, with constant step-sizes leading to stabilization and vanishing step-sizes achieving $O(\log(k)^2/k)$ rates. Future research could refine the analysis by comparing stochastic implicit and iterative differentiation, develop a minibatch version, and explore outcomes in non-strongly convex or nonsmooth settings. Additionally, the feasibility of stochastic iterative differentiation warrants further investigation, given its potential benefits and challenges in such scenarios.

## Acknowledgements

The authors acknowledge the support of ANR MAD ANR-24-CE23-1529. Edouard P. and Franck I. are supported by the AI Interdisciplinary Institute ANITI (ANR-19-PI3A-0004) and ANR REGULIA. Edouard P. acknowledges the support of the Air Force Office of Scientific Research FA8655-22-1-7012, and TSE partnership. Samuel V. is supported by the AI Interdisciplinary Institute 3IA Côte d'Azur (ANR-19-P3IA-0002) and ANR GraVa ANR-18-CE40-0005.

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

# A  Justification of the permutation of integrals and derivatives

We may assume without loss of generality that both $f(0, 0; \xi)$ and $\nabla_{(x,\theta)} f(0, 0; \xi)$ are integrable thanks to Assumption 2(b). Concatenate the variables $x$ and $\theta$, such that $z = (x, \theta)$ and consider the function

$$g \colon (z; \xi) \mapsto \frac{f(z; \xi)}{\|z\|^2 + 1}.$$

Since the gradient of $f$ is $L$-Lipschitz in $z$ by Assumption 1(b), we have using the descent lemma (Nesterov, 2013, Lemma 1.2.3)

$$|f(z; \xi) - f(0; \xi)| \le \|\nabla_z f(0; \xi)\| \|z\| + \frac{L}{2} \|z\|^2$$

so that $g$ is upper bounded by an integrable function uniformly in $z$ as

$$|g(z; \xi)| \le |f(0; \xi)| + \|\nabla_z f(0; \xi)\| + \frac{L}{2}. \tag{5}$$

We also have

$$\nabla_z g(z; \xi) = \nabla_z f(z; \xi) \frac{1}{\|z\|^2 + 1} - z \frac{2f(z; \xi)}{(\|z\|^2 + 1)^2} = \nabla_z f(z; \xi) \frac{1}{\|z\|^2 + 1} - z \frac{2g(z; \xi)}{\|z\|^2 + 1}$$

$$= \nabla_z f(0; \xi) \frac{1}{\|z\|^2 + 1} + (\nabla_z f(z; \xi) - \nabla_z f(0; \xi)) \frac{1}{\|z\|^2 + 1} - z \frac{2g(z; \xi)}{\|z\|^2 + 1}$$

Using again Lipschitz continuity of the gradient of $f$, $\nabla_z g(z; \xi)$ is upper bounded by an integrable function, uniformly in $z$, as

$$\|\nabla_z g(z; \xi)\| \le \|\nabla_z f(0; \xi)\| + L + 2g(z; \xi) \tag{6}$$
$$\le 3\|\nabla_z f(0; \xi)\| + 2L + 2|f(0; \xi)|.$$

Hence, we have that i) $\nabla_z g(z; \xi)$ exists for all $z$ (as $f$ is $C^1$) and ii) both $\xi \mapsto g(z; \xi)$ and $\xi \mapsto \nabla_z g(z; \xi)$ are bounded by functions in $L^1(\mathrm{P})$ uniformly in $z$ thanks to (5) and (6) since $|f(0; \xi)|$ and $\|\nabla_z f(0; \xi)\|$ belong to $L^1(\mathrm{P})$. Hence, we have the appropriate domination assumptions to differentiate under the integral for the function $g$ so that for all $z$, the function $G : z \mapsto \mathbb{E}[g(z; \xi)]$ is differentiable and $\nabla_z G(z) = \mathbb{E}[\nabla_z g(z; \xi)]$ (see e.g., (Folland, 1999, Th. 2.27)).

Now, turning back to $f$, since for all $z$, $f(z; \xi) = g(z; \xi)(\|z\|^2 + 1)$, $F(z) = G(z)(\|z\|^2 + 1)$ and thus $\nabla_z F(z) = \nabla_z G(z)(\|z\|^2 + 1) + 2zG(z)$. Also, for all $z$

$$\nabla_z f(z; \xi) = \nabla_z g(z; \xi)(\|z\|^2 + 1) + 2zg(z; \xi)$$

whose right hand side is integrable as shown above. This enables us to conclude that for all $z$,

$$\mathbb{E}[\nabla_z f(z; \xi)] = \mathbb{E}[\nabla_z g(z; \xi)](\|z\|^2 + 1) + 2z\mathbb{E}[g(z; \xi)]$$
$$= \nabla_z G(z)(\|z\|^2 + 1) + 2zG(z) = \nabla_z F(z).$$

As for the second derivative, $\nabla_z f(z; \xi)$ is $C^1$ with uniformly bounded derivatives so that we may apply differentiation under the integral once again to obtain that the Hessian of the expectation is the expectation of the Hessian.

# B  Proofs from the main text

## B.1  Proof of Lemma 2.1

**Lemma B.1.** *Let $\mu > 0$ and $(\eta_k)_{k \in \mathbb{N}}$ be a sequence of positive numbers. Assume that $(D_k)_{k \in \mathbb{N}}$ is a sequence of matrices of fixed size, such that $D_{k+1} = A_k D_k + B_k$, for matrices $(A_k)_{k \in \mathbb{N}}$ and $(B_k)_{k \in \mathbb{N}}$ of appropriate size where $\|A_k\|_{\mathrm{op}} \le 1 - \mu\eta_k$ and $\|B_k\| \le B\eta_k$ for all $k$. Then for all $k$, $\|D_k\| \le \max\{\|D_0\|, B/\mu\}$.*

*Proof.* We have, using the fact that $\|A_k D_k\| \leq \|A_k\|_{\text{op}} \|D_k\|$,

$$\|D_{k+1}\| = \|A_k D_k + B\| \leq \|A_k D_k\| + \|B_k\| \leq \|A_k\|_{\text{op}} \cdot \|D_k\| + \|B_k\| \leq (1 - \mu\eta_k)\|D_k\| + B\eta_k.$$

There are two cases.

- If $\|D_k\| \geq B/\mu$, then $\|D_{k+1}\| \leq (1 - \mu\eta_k)\|D_k\| + \eta_k B \leq \|D_k\|$.

- If $\|D_k\| \leq B/\mu$, then $\|D_{k+1}\| \leq (1 - \mu\eta_k)B/\mu + \eta_k B = B/\mu$.

The proof is then by induction: if $\|D_k\| \leq \max\{\|D_0\|, B/\mu\}$, the property extends to $D_{k+1}$ by using one of the two cases. $\qquad\square$

*Proof of Lemma 2.1.* The recursion (SGD') can be written

$$D_{k+1} = A_k D_k + B_k$$

where for all $k$, $D_k = \partial_\theta x_k(\theta)$, $A_k = I - \eta_k \nabla^2_{xx} f(x_k(\theta), \theta; \xi_{k+1})$ and $B_k = -\eta_k \nabla^2_{x\theta} f(x_k(\theta), \theta; \xi_{k+1})$. Using Assumption 1, we have that $\|A_k\|_{\text{op}} \leq 1 - \eta_k\mu$ and $\|B_k\| \leq \sqrt{m}\|B_k\|_{\text{op}} \leq \sqrt{m}L'\eta_k$. The result follows from Lemma B.1. $\qquad\square$

## B.2  Proof of Proposition 3.1

*Proof of Proposition 3.1.* First, we recall that the expected norm of a stochastic gradient can be controlled for any $k \in \mathbb{N}$ as

$$\mathbb{E}\big[\|\nabla_x g(x_k; \xi_{k+1})\|^2 | \mathcal{F}_k\big] \leq 2\mathbb{E}\big[\|\nabla_x g(x^\star; \xi_{k+1})\|^2 | \mathcal{F}_k\big] + 2\mathbb{E}\big[\|\nabla_x g(x_k; \xi_{k+1}) - \nabla_x g(x^\star; \xi_{k+1})\|^2 | \mathcal{F}_k\big]$$
$$\leq 2\sigma^2 + 2L^2\|x_k - x^\star\|^2 \tag{7}$$

where we used Assumption 3($a$) and ($c$) in the second inequality.

By definition of (2), we have for all $k \in \mathbb{N}$

$$\|x_{k+1} - x^\star\|^2 = \|x_k - x^\star\|^2 + \eta_k^2\|\nabla_x g(x_k; \xi_{k+1}) + e_{k+1}\|^2 - 2\eta_k\langle x_k - x^\star, \nabla_x g(x_k; \xi_{k+1}) + e_{k+1}\rangle$$
$$\leq \|x_k - x^\star\|^2 + 2\eta_k^2\big(\|\nabla_x g(x_k; \xi_{k+1})\|^2 + \|e_{k+1}\|^2\big) - 2\eta_k\langle x_k - x^\star, \nabla_x g(x_k; \xi_{k+1})\rangle$$
$$+ 2\eta_k\|x_k - x^\star\|\|e_{k+1}\|.$$

Taking the expectation conditioned on $\mathcal{F}_k$, we get with our assumption on the errors that

$$\mathbb{E}\big[\|x_{k+1} - x^\star\|^2 | \mathcal{F}_k\big] \leq \|x_k - x^\star\|^2 + \eta_k^2\big(4L^2\|x_k - x^\star\|^2 + 4\sigma^2 + 2\mathbb{E}\big[\|e_{k+1}\|^2 | \mathcal{F}_k\big]\big)$$
$$- 2\eta_k\langle x_k - x^\star, \mathbb{E}[\nabla_x g(x_k; \xi_{k+1}) | \mathcal{F}_k]\rangle$$
$$+ 2\eta_k\|x_k - x^\star\|\mathbb{E}[\|e_{k+1}\| | \mathcal{F}_k]$$
$$\leq \big(1 - 2\eta_k\mu + 4\eta_k^2 L^2\big)\|x_k - x^\star\|^2 + \eta_k^2\big(4\sigma^2 + 2\mathbb{E}\big[\|e_{k+1}\|^2 | \mathcal{F}_k\big]\big)$$
$$+ 2\eta_k\|x_k - x^\star\|\mathbb{E}[\|e_{k+1}\| | \mathcal{F}_k] \tag{8}$$

where we used successively Eq. (7) and Assumption 3($b$). Now using Jensen's inequality and the Cauchy-Schwartz inequality: $\mathbb{E}[XY] \leq \sqrt{\mathbb{E}[X^2]\mathbb{E}[Y^2]}$ for square integrable random variables, we have the following bound on the full expectation of the last product,

$$\mathbb{E}[\|x_k - x^\star\|\mathbb{E}[\|e_{k+1}\| | \mathcal{F}_k]] \leq \sqrt{\mathbb{E}[\|x_k - x^\star\|^2]\mathbb{E}\big[\mathbb{E}[\|e_{k+1}\| | \mathcal{F}_k]^2\big]}$$
$$\leq \sqrt{\mathbb{E}[\|x_k - x^\star\|^2]}\sqrt{\mathbb{E}[\mathbb{E}[\|e_{k+1}\|^2 | \mathcal{F}_k]]}$$
$$= \sqrt{\mathbb{E}[\|x_k - x^\star\|^2]}\sqrt{\mathbb{E}[\|e_{k+1}\|^2]}$$

Now, our condition on the stepsize parameters implies that $-2\eta_k\mu + 4\eta_k^2 L^2 \leq -\eta_k\mu$. By taking full expectation on both sides of (8), we obtain that

$$\mathbb{E}\big[\|x_{k+1} - x^\star\|^2\big] \leq (1 - \eta_k\mu)\mathbb{E}\big[\|x_k - x^\star\|^2\big] + \eta_k^2\big(4\sigma^2 + 2B_k^2\big) + 2\eta_k\sqrt{\mathbb{E}[\|x_k - x^\star\|^2]}B_k$$

We set $D_k = \sqrt{\mathbb{E}[\|x_k - x^\star\|^2]}$ so that we have the following deterministic recursion:

$$D_{k+1}^2 \leq (1 - \mu\eta_k)D_k^2 + 2\eta_k^2(B_k^2 + 2\sigma^2) + 2\eta_k B_k D_k.$$

$\qquad\square$

## C    Technical Lemmas

**Lemma C.1.** *Let $(\eta_k)_{k\in\mathbb{N}}$ and $(B_k)_{k\in\mathbb{N}}$ be non-negative and non-increasing. Assume that $(\eta_k)_{k\in\mathbb{N}}$ is non-summable and that $0 < \eta_k \le \frac{1}{\mu}$ for all $k$. Let $(D_k)_{k\in\mathbb{N}}$ be a non-negative sequence satisfying for all $k$*

$$D_{k+1}^2 \le (1 - \mu\eta_k) D_k^2 + 2\eta_k^2(B_k^2 + 2\sigma^2) + 2\eta_k B_k D_k . \tag{9}$$

*Consider the quantity*

$$\delta_k = \frac{\sqrt{4\eta_k^2 B_k^2 + 8\mu\eta_k^3(B_k^2 + 2\sigma^2)} + 2B_k\eta_k}{2\mu\eta_k} = \frac{\sqrt{B_k^2 + 2\mu\eta_k(B_k^2 + 2\sigma^2)} + B_k}{\mu}.$$

*Then, $(\delta_k)_{k\in\mathbb{N}}$ is positive, non-increasing, and for any $\delta > \lim_{k\to\infty}\delta_k$*

$$\limsup_{k\to\infty} \quad D_k \le \delta.$$

*Proof.* Set for each $k \in \mathbb{N}$, $F_k \colon \mathbb{R}_+ \to \mathbb{R}_+$, with $F_k(t) = (1 - \mu\eta_k)\, t + 2\eta_k B_k \sqrt{t} + 2\eta_k^2(B_k^2 + 2\sigma^2)$. We have that $F_k$ is increasing, concave, and $F_k(\delta_k^2) = \delta_k^2$. By assumption, for all $k$ sufficiently large, we have $\delta_k < \delta$ so that $F_k(\delta^2) \le \delta^2$ as $t \mapsto F_k(t^2) - t^2$ is negative for $t \ge \delta_k$.

Plugging this into (9), we obtain

$$D_{k+1}^2 - \delta^2 \le (1 - \mu\eta_k) D_k^2 + 2\eta_k B_k D_k + 2\eta_k^2(B_k^2 + 2\sigma^2) - F_k(\delta^2)$$
$$= (1 - \mu\eta_k)(D_k^2 - \delta^2) + 2\eta_k B_k(D_k - \delta).$$

Using the fact that $\mu\eta_k \le 1$, we deduce that if $D_k \le \delta$, then $D_{k+i} \le \delta$ for all $i \in \mathbb{N}$ and the result follows. We continue assuming that $D_k > \delta$ for all $k \in \mathbb{N}$.

Using the concavity of the square root, we have $D_k - \delta = \sqrt{D_k^2} - \sqrt{\delta^2} \le \frac{1}{2\sqrt{\delta^2}}(D_k^2 - \delta^2)$. We deduce that

$$D_{k+1}^2 - \delta^2 \le \left(1 - \mu\eta_k + \frac{\eta_k B_k}{\delta}\right)(D_k^2 - \delta^2).$$

We notice that for all $k$, $\frac{2B_k}{\mu} \le \delta_k$ so that for $k$ large enough, $\frac{2B_k}{\mu} \le \delta$, and $\frac{\eta_k B_k}{\delta} \le \frac{\mu\eta_k}{2}$, and we obtain

$$D_{k+1}^2 - \delta^2 \le \left(1 - \frac{\mu\eta_k}{2}\right)(D_k^2 - \delta^2).$$

So there is an index $k_0$ such that for all $k \ge k_0$, we have $D_k^2 - \delta^2 \le \prod_{i=k_0}^k \left(1 - \frac{\mu\eta_i}{2}\right)(D_{k_0}^2 - \delta^2)$ and the right hand side decreases to 0 as $k \to \infty$ because $\eta_k$ is non-summable. This concludes the proof. $\qquad\square$

**Lemma C.2.** *Let $\eta_k = \frac{2\mu}{\mu^2 k + 8L^2}$ for all $k \in \mathbb{N}$ and $(D_k)_{k\in\mathbb{N}}$ be a non-negative sequence satisfying, for all $k$,*

$$D_{k+1}^2 \le (1 - \mu\eta_k) D_k^2 + 4\eta_k^2 \sigma^2.$$

*Then we have, for all $k \in \mathbb{N}$,*

$$D_{k+1}^2 \le \frac{1}{k + 8\kappa^2}\left(8\kappa^2 D_0^2 + \frac{2\sigma^2}{L^2} + \frac{16\sigma^2}{\mu^2}\log\left(1 + \frac{k}{8\kappa^2}\right)\right).$$

*Proof.* From the recursion, we obtain

$$D_{k+1}^2 \le \left(1 - \frac{2\mu^2}{\mu^2 k + 8L^2}\right) D_k^2 + \frac{16\mu^2\sigma^2}{(\mu^2 k + 8L^2)^2}$$

$$(\mu^2 k + 8L^2)D_{k+1}^2 \le (\mu^2 k + 8L^2 - 2\mu^2) D_k^2 + \frac{16\mu^2\sigma^2}{(\mu^2 k + 8L^2)}$$

$$\le (\mu^2(k-1) + 8L^2) D_k^2 + \frac{16\mu^2\sigma^2}{(\mu^2 k + 8L^2)}$$

from which we deduce that

$$(\mu^2 k + 8L^2)D_{k+1}^2 \leq (8L^2 - \mu^2)\, D_0^2 + \sum_{i=0}^{k} \frac{16\mu^2\sigma^2}{(\mu^2 i + 8L^2)}$$

$$\leq 8L^2 D_0^2 + 16\sigma^2 \sum_{i=0}^{k} \frac{1}{(i + \frac{8L^2}{\mu^2})}$$

$$\leq 8L^2 D_0^2 + 16\sigma^2 \left( \frac{\mu^2}{8L^2} + \log\left(1 + \frac{k\mu^2}{8L^2}\right) \right)$$

where the last inequality is by integral series comparison. All in all, we obtain

$$D_{k+1}^2 \leq \frac{8L^2 D_0^2}{\mu^2 k + 8L^2} + \frac{16\sigma^2}{\mu^2 k + 8L^2}\left( \frac{\mu^2}{8L^2} + \log\left(1 + \frac{k\mu^2}{8L^2}\right) \right)$$

$$= \frac{8\kappa^2 D_0^2}{8\kappa^2 + k} + \frac{2\sigma^2}{L^2(k + 8\kappa^2)} + \frac{16\sigma^2 \log\left(1 + \frac{k\mu^2}{8L^2}\right)}{\mu^2(k + 8\kappa^2)}$$

$$= \frac{1}{k + 8\kappa^2}\left( 8\kappa^2 D_0^2 + \frac{2\sigma^2}{L^2} + \frac{16\sigma^2}{\mu^2}\log\left(1 + \frac{k}{8\kappa^2}\right) \right).$$

$\square$

**Lemma C.3.** *Let* $\eta_k = \frac{2\mu}{\mu^2 k + 8L^2}$, *for all* $k \in \mathbb{N}$, $\kappa = \frac{L}{\mu}$, *and* $(D_k)_{k\in\mathbb{N}}$ *be a non-negative sequence satisfying, for all* $k$,

$$D_{k+1}^2 \leq (1 - \mu\eta_k)\, D_k^2 + 2\eta_k^2(B_k^2 + 2\sigma^2) + 2\eta_k B_k D_k\,.$$

*where there are constants* $A, B > 0$ *such that, for all* $k \in \mathbb{N}$,

$$B_k^2 \leq \frac{A + B\log\left(k + 8\kappa^2\right)}{k + 8\kappa^2}.$$

*Then, we have*

$$D_{k+1}^2 \leq \frac{8\kappa^2 D_0^2}{k + 8\kappa^2} + \frac{1}{\mu^2}\frac{\left(5(B + A) + 8\sigma^2\right)\log(k + 8\kappa^2)^2}{k + 8\kappa^2}$$

*Proof.* We first rework the recursion, we use the fact that

$$2\eta_k B_k D_k = 2\eta_k \left( \frac{\sqrt{2}B_k}{\sqrt{\mu}} \right)\left( \frac{\sqrt{\mu}}{\sqrt{2}}D_k \right) \leq \eta_k \left( \frac{2B_k^2}{\mu} + \frac{\mu}{2}D_k^2 \right) = \frac{2\eta_k B_k^2}{\mu} + \eta_k \frac{\mu}{2}D_k^2\,.$$

The new recursion becomes

$$D_{k+1}^2 \leq \left(1 - \frac{\mu\eta_k}{2}\right) D_k^2 + 2\eta_k^2(B_k^2 + 2\sigma^2) + \frac{2\eta_k B_k^2}{\mu}\,. \tag{10}$$

From this recursion, we obtain by expanding all terms

$$D_{k+1}^2 \leq \left(1 - \frac{\mu^2}{\mu^2 k + 8L^2}\right) D_k^2 + \frac{8\mu^2}{(\mu^2 k + 8L^2)^2}\left( 2\sigma^2 + \frac{A + B\log\left(k + 4\kappa^2\right)}{k + 4\kappa^2} \right)$$

$$+ \frac{2\mu}{(\mu^2 k + 8L^2)}\frac{2(A + B\log\left(k + 8\kappa^2\right))}{\mu(k + 8\kappa^2)}$$

$$(\mu^2 k + 8L^2)D_{k+1}^2 \leq (\mu^2 k + 8L^2 - \mu^2)\, D_k^2 + \frac{8}{(k + 8\kappa^2)}\left( 2\sigma^2 + \frac{(A + B\log\left(k + 8\kappa^2\right))}{(k + 8\kappa^2)} \right)$$

$$+ \frac{4(A + B\log\left(k + 8\kappa^2\right))}{(k + 8\kappa^2)}$$

$$\leq (\mu^2(k - 1) + 8L^2)\, D_k^2 + \frac{\log\left(k + 8\kappa^2\right)}{k + 8\kappa^2}\left(5(B + A) + 16\sigma^2\right)$$

where we use the fact that $k \geq 0$ and $\kappa \geq 1$ so that $\log\left(k + 8\kappa^2\right) \geq \log\left(8\right) > 1$. We deduce that

$$(\mu^2 k + 8L^2)D_{k+1}^2 \leq \left(8L^2 - \mu^2\right)D_0^2 + \left(5(B + A) + 16\sigma^2\right)\sum_{i=0}^{k}\frac{\log(i + 8\kappa^2)}{(i + 8\kappa^2)}$$

$$\leq 8L^2 D_0^2 + \left(5(B + A) + 16\sigma^2\right)\log(k + 8\kappa^2)^2$$

where the last inequality is by integral series comparison, using the fact that $t \mapsto \log(t)/t$ is decreasing for $t \geq \exp(1)$, we have

$$\sum_{i=0}^{k}\frac{\log(i + 8\kappa^2)}{(i + 8\kappa^2)} \leq \frac{\log(8\kappa^2)}{8\kappa^2} + \log(k + 8\kappa^2)^2 - \log(8\kappa^2)^2 \leq \log(k + 8\kappa^2)^2.$$

$\square$

**Lemma C.4.** *Let $\eta_k = \eta < \frac{1}{2\mu}$ for all $k \in \mathbb{N}$, $\kappa = \frac{L}{\mu}$, and $(D_k)_{k\in\mathbb{N}}$ be a non-negative sequence satisfying for all $k$*

$$D_{k+1}^2 \leq (1 - \mu\eta_k)D_k^2 + 2\eta_k^2 B_k^2 + 2\eta_k B_k D_k.$$

*where, there is a constant $A > 0$, with $\rho = 1 - \frac{\mu\eta}{2}$ such that, for all $k \in \mathbb{N}$,*

$$B_k^2 \leq A\rho^k.$$

*Then, we have*

$$D_k^2 \leq \rho^k\left(D_0^2 + \frac{kA}{\rho}\left(2\eta^2 + 2\frac{\eta}{\mu}\right)\right).$$

*Proof.* We proceed similarly as in (10) and obtain

$$D_{k+1}^2 \leq \left(1 - \frac{\mu\eta_k}{2}\right)D_k^2 + 2\eta_k^2 B_k^2 + \frac{2\eta_k B_k^2}{\mu} \leq \rho D_k^2 + A\rho^k\left(2\eta^2 + 2\frac{\eta}{\mu}\right).$$

We rewrite and use an induction to obtain

$$\frac{D_{k+1}^2}{\rho^{k+1}} \leq \frac{D_k^2}{\rho^k} + \frac{A}{\rho}\left(2\eta^2 + 2\frac{\eta}{\mu}\right) \leq D_0^2 + \frac{kA}{\rho}\left(2\eta^2 + 2\frac{\eta}{\mu}\right)$$

which is the desired result. $\square$

