# OpenReview forum: "Derivatives of Stochastic Gradient Descent in parametric optimization"
_NeurIPS.cc/2024/Conference — NeurIPS 2024 poster_

### Official Review · Reviewer_cZZE · 2024-06-25

**Soundness:** 3
**Presentation:** 3
**Contribution:** 2
**Rating:** 6
**Confidence:** 1

**Summary:**

The paper studies stochastic optimization problems where the objective depends on a parameter, and more specifically the derivatives w.r.t. that parameter of the SGD iterates. The paper makes various quite strong albeit common assumptions, and for various more specific settings concrete convergence rates are established. The proof relies on analysis via an inexact SGD sequence. Finally, the results are illustrated by some experiments.

**Strengths:**

The paper is well-written and the technical flow of ideas is convincing. Especially the connection to inexact SGD seems quite novel.

**Weaknesses:**

My main concern is of a motivational nature: why is the question set out inn ll. 35-36 interesting? I do not find ll. 37 sufficiently convincing. (I may not have understood that paragraph fully.)

**Questions:**

1. Can you elaborate on the motivation?
2. Why do you need that the gradient for the first iterates vanish (l. 194)? That appears to be an odd and strong assumption.
3. How restrictive are the assumptions? (I understand that they are mostly common in the literature.)

**Limitations:**

Assumptions are discussed.

---

> ### Author Rebuttal · Authors · 2024-08-02
>
> **Motivation:** We made a common response to all reviewers regarding the motivation. We will modify this section to include explicit references to works which consider differentiating SGD sequences or propose it as a relevent research venue.
>
> **Vanishing derivative at initialization:** this remark was also made by reviwer **bPEV**, we will remove this assumption. This choice was only  made for simplicity to avoid  the term $\|\partial_\theta x_0(\theta)\|$ in the right hand side of the estimate in Lemma 2.1. This will be corrected in the revision, we will remove this assumption and explicit all the terms, including a dependency in the derivative of the initialization. This represents a very minor modification.
>
> **Discussion of the assumptions:** The crucial assumption for our results is strong convexity. The rest of the assumptions are typically satisfied in applications such as hyper parameter tuning. We point out that both examples in the numerical section satisfy our assumtions and are implemented in the regime described by our main theorem. Our assumptions are classical, as described for example in the two general optimization references presented in the response to all reviewers regarding our step size choice. We will add this discussion in a revised version of the paper.

---

> ### Author Response · Authors · 2024-08-11
>
> Dear Reviewer cZZE,
>
> Thank you again for your detailed feedback on our paper. We hope that our rebuttal addressed the concerns you raised. If you have any further questions or require additional clarifications, we would be happy to provide them.
>
> If you are satisfied with our responses, we kindly ask you _to consider_ raising your score in the light of our responses. We appreciate your time and effort in reviewing our work.
>
> Best regards,
>
> Authors

---

> ### Comment · Reviewer_cZZE · 2024-08-12
>
> Thank you for your clarifications. The authors have addressed my points. In particular, I belief that the authors will be able to present a more convincing case for their motivation in a revision.
>
> I belief my original assessment of the paper's impact is accurate.

---

### Official Review · Reviewer_bPEV · 2024-07-04

**Soundness:** 3
**Presentation:** 3
**Contribution:** 3
**Rating:** 7
**Confidence:** 4

**Summary:**

The authors consider parametric stochastic optimization problems of the form $\min_x F(\theta,x)$ where $F(\theta,x)= \mathbb E_\xi [ f(x, \theta, \xi)]$ under the condition that $f$ is strongly convex in $x$ for any fixed $\theta, \xi$. This ensures that for fixed $\theta$, there exists a unique minimizer $x^*(\theta)$. The authors construct a sequence $x_k$ by gradient descent for $F(\theta, \cdot)$ and demonstrate that an associated sequence, denoted by $\partial_\theta x_k$, converges to the parameter derivative $\partial_\theta x^*(\theta)$ of the solution map $x^*(\theta)$. They explore various regimes of stochasticity and various learning rate schedules, obtaining rates in different settings. Strong continuity assumptions are placed on $f, \nabla f, D^2f$.

**Strengths:**

The article is generally well-written and explains its results intuitively. The statements of results are precise, yet clear.

**Weaknesses:**

* I find the title almost misleading. The authors do not consider derivatives (modifications) of the algorithm, but they consider the derivatives of iterates with respect to an additional parameter. I would propose something along the lines of "Derivatives in parametric optimization and their behavior along SGD iteration".

* I do not follow the numerical illustration in Section 4 at all. The independent parameter $\theta$ becomes a random quantity in this section, and it is unclear what stochastic gradient estimates the authors use: Only the deterministic objective function $F$ is specified (line 291), and $\xi$ is never mentioned in this section. I am not sure why the authors draw $\theta$ from a random distribution. I believe that the experiments either do not match the setting considered above or the presentation needs to be clarified considerably.

**Questions:**

* Is there a conflict between the statement in line 131 that "[...] the initialization of the algorithm $x_0(\theta)$ depend[s] on some parameter $\theta$" and the assumption that $\partial_\theta x_0(\theta) =0$ in Theorem 2.2?

* In Lemma 2.1, the restriction that $\eta_k \leq \mu/L^2$ is generally much more severe than the usual bound $\eta_k \leq 1/L$. Perhaps by considering the quadratic case, could the authors speculate whether it is necessary to ensure the convergence of derivatives or whether it could be relaxed?

* In Remark 2.3, if we use a lower estimate for $\mu$, this also changes the estimate for $\kappa$: The constants $c, u$ are not independent. In fact, $\eta_0 = 1/(4\mu \kappa^2) = \mu/(4L^2)$ satisfies the necessary condition above. If we increased $c$ without adjusting $u$, we may easily enter a regime where $\eta_0> \mu/L^2$. A more careful consideration appears to be needed.

**Limitations:**

Yes

---

> ### Author Rebuttal · Authors · 2024-08-02
>
> **Title:** We propose to add ''in parametric optimization'' at the end of the title if you believe it better illustrates our results. An alternative would be to name it "Derivatives through Stochastic Gradient Descent".
>
> **Numerical section:** Thanks for pointing out this possible confusion, we will provide a more precise description of the generation of synthetic problem and explicit the stochastic setting as follows:
> - The randomness of $\theta$ corresponds to the choice of a specific $\theta$ and is secondary in the experiment. The dependency in $\theta$ of the objective through $b(\theta)$ is the most important one and we will put emphasis on this rather than the choice of a fixed $\theta$. In its current state, the presentation is indeed misleading
> - The least squares objective has the structure of a finite sum and the randomness in SGD is the classical with replacement sampling, which fits our assumptions. The same comment holds for the second example. We will make this precise in the revision.
>
> **Dependency on initial parameters** This remark was also made by reviewer **cZZE**. Indeed, we assume no dependency of the initialization on parameters $\theta$. This choice was only  made for simplicity to avoid the term $\|\partial_\theta x_0(\theta)\|$ in the right hand side of the estimate in Lemma 2.1. This will be corrected in the revision, we will remove this assumption and explicit all the terms, including a dependency in the derivative of the initialization. This represents a very minor modification but should clarify the concern of the reviewer.
>
> **Step size condition:** We made a common response to all reviewers regarding this restriction. We do not believe that our analysis is optimal, but conjecture that the dependency on $\mu$ is required. This is mostly related to the need to consider the specific strongly convex regime for which it is classical to require strong conditions on the step size.
>
> **Remark 2.3:** The reviewer is right, this is presented in an incorrect way. The remark will be modified as follows:
>
> The specific stepsize used to obtain the sublinear rate actually applies to any stepsize of the form $\eta_k = 2/(c k+8u)$ for given $c,u>0$ such that $0<c\leq \mu$ and $u \geq L^2/c$. One obtains the same result with $\mu, L$ respectively replaced in the expressions by $\mu' :=  c \leq \mu$ and $L' := \sqrt{ u c} \geq  L$.
>
> We would like to kindly ask the reviewer to take into account these responses and possibly reconsider his evaluation of our work.

---

> > ### Comment · Reviewer_bPEV · 2024-08-08
> >
> > Thank you for the comprehensive answer. I believe that with minor corrections, my concerns can be addressed. I choose to raise my score to 7.
> >
> > For strongly convex optimization, it should be noted that in general the learning rate is $1/L$, not $\mu/L^2$ when considering the convergence of the objective function rather than its derivatives. This is true for both gradient descent and Nesterov's method. Naturally, smaller step sizes have to be chosen in the stochastic case. I would be curious to see a broader exploration.

---

> > > ### Author Response · Authors · 2024-08-11
> > >
> > > Dear reviewer bPEV,
> > >
> > > Thank you very much for your thoughtful feedback and for raising your score!  You bring up an excellent point regarding the learning rate in the context of strongly convex optimization. Indeed, the distinction you mention between the convergence rates of the objective function and its derivatives is important. Exploring this aspect in more detail is indeed an interesting direction for future work.
> > >
> > > Best,
> > >
> > > Authors

---

### Official Review · Reviewer_d938 · 2024-07-11

**Soundness:** 3
**Presentation:** 3
**Contribution:** 3
**Rating:** 4
**Confidence:** 3

**Summary:**

The paper considers stochastic optimizations where the objective depends on some parameter.  Instead of the SGD, the paper considers the derivatives of the iterates of the SGD with respect to that parameter in the context where the objective is strongly convex.
Convergence analysis is obtained for the derivatives of the iterates of SGD, which can be viewed as an inexact SGD on a different objective, perturbed by the convergence of the original SGD.

**Strengths:**

(1) Convergence guarantees are obtained for derivatives of SGD under certain assumptions.

(2) Analysis seems to be rigorous and solid.

**Weaknesses:**

(1) Derivatives of SGD are much less studied than SGD. As a result, it would be helpful to add more discussions about Assumption 1, Assumption 2, what kind of examples of interest satisfy these two assumptions (especially the part that is unique to the setting that involves the parameter).

(2) Numerical experiments are only synthetic. It would be nice if the paper can add an experiment on real data.

(3) I do not see adequate discussions whether the assumptions for the theoretical part can be satisfied for the examples considered
in the numerical section.

(4) Inexact SGD has been well studied in the literature. Since the paper views derivatives of SGD as an inexact SGD, it is not clear
what technical novelty and contributions arise from this context.

**Questions:**

(1) In Theorem 2.2, I understand that constant step-size is a popular choice in the SGD literature. But can you comment on the choice $\eta_{k}=\frac{1}{\mu}\frac{2}{k+8\kappa^{2}}$ for the sublinear rate regime, and moreover the interpolation regime, in which the assumption $\sigma=0$ seems to be super strong to me, and can you provide some examples of interest that satisfy this particular assumption?

(2) In the paragraph after equation (1), you wrote that the error term is of order... Please specify which term is the error term.

(3) To improve the readability of the paper, I suggest you state somewhere in the main paper how you view the derivatives of SGD as an inexact SGD. For example, in the proof of Theorem 2.2., you defined $e_{k+1}$, and I think you can define $e_{k+1}$, as well as $\nabla_{x}g(x_{k};\xi_{k+1})$ when you explain how you view the derivatives of SGD as an inexact SGD before you state the main results to help the readers understand better.

(4) Some of the journal names in the references should be capitalized. For example, for Robbins and Monro,
it should be The Annals of Mathematical Statistics.

**Limitations:**

More discussions on limitations should be added.

---

> ### Author Rebuttal · Authors · 2024-08-02
>
> ### Weaknesses:
>
>
> **(1) Discussion of the assumptions:** The crucial assumption for our results is strong convexity. The rest of the assumptions are typically satisfied in applications such as hyper parameter tuning. We point out that both examples in the numerical section satisfy our assumtions and are implemented in the regime described by our main theorem. The rest of the assumptions are classical, as described for example in the two general optimization references presented in the response to all reviewers regarding our step size choice. We will add this discussion in a revised version of the paper.
>
>
> **(2) Real data:** We will add experiments using the same models as in the numerical section, with real data, in the appendix. We provide in the attached PDF on OpenReview a numerical experiment for the derivatives of SGD on a logistic regression problem on the dataset IJCNN1 (~50k samples). The iterates of SGD are differentiated with respect to the L2 regularization parameter of the logistic regression loss. The observed behavior is very to close to the one observed in the synthetic experiments (see Fig. 2 of the paper), validating our approach. Note that non-strongly convex behaviours is out-of-the-scope of this paper, since we provide the theory necessary to study the strongly convex case.
>
> **(3) Numerical section:** Both examples in the numerical section satisfy our assumtions and are implemented in the regime described by our main theorem. The numerical section will be reworked to state this explicitely.
>
> **(4) Literature on inexact SGD:** Indeed, the paper misses this element. We base our discussion on the recent publication:
> - *Demidovich, Malinovsky, Sokolov, Richtárik. A guide through the zoo of biased sgd. Neurips 2023*.
>
> We provide a general mean squared error convergence analysis of inexact SGD which allows to handle random non stationary bias terms, whose magnitude depend on the iteration counter $k$. This is customary as our errors depend on the realization of the SGD iterate sequence, requiring a dedicated analysis not covered by existing literature on inexact SGD. We will add a paragraph about this discussion at the end of the introduction.
>
> ### Questions:
>
> **(1) Step size and interpolation** Please see our common response to all reviewers regarding the choice of step size. The case $\sigma = 0$ is often used in optimization and ML literature as an idealized model to capture overparametrization with very large networks. It is indeed a very strong condition (note though that the absence of noise is _only_ true at the optimum). Such an interpolation regime was studied in various works, including (but not limited to):
> - *Ma, Bassily, Belkin, (2018). The power of interpolation: Understanding the effectiveness of SGD in modern over-parametrized learning. ICML*. This paper studies linear convergence of SGD in the interpolation regime.
> - *Varre, A. V., Pillaud-Vivien, L., & Flammarion, N. (2021). Last iterate convergence of SGD for Least-Squares in the Interpolation regime. Neurips*. This paper studies interpolation at the population level for linear regression.
> - *Garrigos, G., & Gower, R. M. (2023). Handbook of convergence theorems for (stochastic) gradient methods. arXiv preprint*. The interpolation regime is mentioned in several places in this manuscript.
>
> Although this looks like an edge case, we included this interpolation regime in the paper because we believe that it is of interest to the ML/OPT community and also because the theory sometimes predicts an surprising behavior: linear convergence of iterates, but not of derivatives, which we thought was interesting.
>
>
> **(2)** Indeed, thanks for catching this, we will make it explicit in the revision.
>
> **(3)** Indeed we will describe the error and notations right after equation (1).
>
> **(4)** Thanks for catching this, we will correct this.
>
>
> We believe that we brought relevant responses to the legitimate questions of the reviewer and we would like to ask the reviewer to reconsider his evaluation in light of the elements given above.

---

> ### Author Response · Authors · 2024-08-11
>
> Dear Reviewer d938,
>
> Thank you again for your detailed feedback on our paper. We hope that our rebuttal addressed the concerns you raised. If you have any further questions or require additional clarifications, we would be happy to provide them.
>
> If you are satisfied with our responses, we kindly ask you _to consider_ raising your score in the light of our responses since your current rating leans toward a reject. We appreciate your time and effort in reviewing our work.
>
> Best regards,
>
> Authors

---

### Official Review · Reviewer_mZD2 · 2024-07-12

**Soundness:** 3
**Presentation:** 3
**Contribution:** 3
**Rating:** 6
**Confidence:** 3

**Summary:**

This is a theoretical paper on iterative process differentiation. The paper analyzes the behavior of the derivatives of the iterates of SGD (Stochastic Gradient Descent). Based on a set of assumptions, the paper establishes the convergence of the derivatives of SGD and conducts numerical experiments to validate its findings.

**Strengths:**

The highlights of the paper are: (1) revealing that the behavior of the derivatives of the iterates is driven by an inexact/perturbed SGD recursion; (2) illustrating their theory with numerical experiments on synthetic tasks.

**Weaknesses:**

I believe the main weaknesses of the paper are that the assumptions used to establish the theory are too strong; moreover, the practical significance of the theory is not clearly articulated, especially in relation to stochastic hyperparameter optimization.

**Questions:**

The smoothness assumption in Assumption 1(b) requires that the gradient are jointly L-Lipschitz continuous in $x$ and $\theta$ . Can this assumption be relaxed?

**Limitations:**

Not applicable.

---

> ### Author Rebuttal · Authors · 2024-08-02
>
> **About the motivation in stochastic hyperparameter optimization and assumptions:** We made a common response to all reviewers regarding theses two points. We will modify the appropriate paragraphs to include explicit references to works which consider differentiating SGD sequences or propose it as a relevent research venue. We will also pay an extra attention to the reach and necessity of our assumptions.
>
> **Jointly Lipschitz Gradient:**
> > The smoothness assumption in Assumption 1(b) requires that the gradient are jointly L-Lipschitz continuous in $x$ and $\theta$. Can this assumption be relaxed?
>
> Joint gradient Lipschicity is indeed a strong assumption, but its strength is mitigated as follows:
> - $\Theta$ is an arbitrary open Euclidean subset, in particular, this could be a small ball around a given $\bar{\theta}$. In other words we do not require global Lipschicity with respect to $\theta$, only local Lipschicity which is a consequence of the $C^2$ assumption and is expressed here in a quantitative form. Global gradient Lipschicity is only required for the $x$ variable which is typical for the analysis of gradient schemes.
> - We chose the same constant $L$ with respect to both $x$ and $\theta$ for simplicity. We prefered simple assumptions and fewer notations. The Lipschicity with respect to $\theta$ is only used in Lemma 2.1 which is in turn used to obtain estimates on the error in equation (4). These could be separated.
>
> As requested by the reviewer, we will relax this assumption in the revision and separate $L_x$, the Lipschitz constant with respect to $x$ for fixed $\theta$, and $L_\theta$ with respect to $\theta$. This will only incur minor modifications of the estimate of Lemma 2.1, the constant $L_x$ being the crucial one in the rest of the analysis. We will also add a precise remark regarding the fact that the constant $L_\theta$ does not need to be a global Lipschitz constant in $\theta$.
>
>
> We believe that this is a nice improvement to our set of assumptions and would like to ask the reviewer to reconsider his evaluation in light of this discussion.

---

> ### Author Response · Authors · 2024-08-11
>
> Dear Reviewer mZD2,
>
> Thank you again for your detailed feedback on our paper. We hope that our rebuttal addressed the concerns you raised. If you have any further questions or require additional clarifications, we would be happy to provide them.
>
> If you are satisfied with our responses, we kindly ask you _to consider_ raising your score in the light of our responses. We appreciate your time and effort in reviewing our work.
>
> Best regards,
>
> Authors

---

> ### Comment · Reviewer_mZD2 · 2024-08-14
>
> Thanks for the clarification / explanation. Although I have not studied those future improvements in detail, I believe that relaxing the assumptions in this interesting theoretical issue could offer more valuable insights for practical work. For now, I would prefer to maintain my original rating.

---

### Author Rebuttal · Authors · 2024-08-02

Dear AC, dear reviewers,

We are sincerely grateful for your time and input. We reply to each of your questions and comments in a separate point-by-point thread below. We will of course integrate all applicable points in the next revision opportunity. We start with two general comments regarding motivations and step size constraints, which raised questions from several reviewers. We hope that the points developed below provide satisfactory answers to your concerns.

Kind regards,
The authors



## Motivation (common to several reviewers)

Several raised concerns about the motivation for our work. We would like to point four relevant bibliographic references which explicitely mention the idea of differentiation through SGD, among existing literature on the topic.
- *Maclaurin, Duvenaud, Adams (2015). Gradient-based hyperparameter optimization through reversible learning. ICML.* This paper is dedicated to an efficient implementation of reverse automatic differentiation for SGD to evaluate its derivatives. Our theory is about Jacobians, it applies to both forward and reverse mode automatic differentiation.
- *Pedregosa (2016). Hyperparameter optimization with approximate gradient. ICML.* This is an important paper in hyper parameter tuning which explicitely calls for the development of differentiation techniques for stochastic optimization algorithms and motivated many subsequent works in iterative differentiation.
- *Finn, Abbeel, Levine (2017). Model-agnostic meta-learning for fast adaptation of deep networks. ICML.* This is a landmark paper on meta learning which suggests to use differentiation through stochastic first order solvers.
- *Ji, Yang, Liang (2022). Theoretical convergence of multi-step model-agnostic meta-learning. JMLR.* Differentiation through SGD is explicitely described and studied in this reference motivated by meta learning applications.

Since the convergence of the derivatives of SGD is *not considered in the literature*, we believe that the elements above constitute sufficiently important motivation to study it more precisely. We will revise the introduction to let these elements appear more clearly. We kindly ask the reviewers to take these elements into consideration in their evaluation.


## Step size constraints and strong convexity (common to several reviewers)

Several reviewers expressed concerns regarding the fact that we do not have the usual $1/L$ step size limitation, but rather the smaller $\mu/L^2$. We emphasize that our study takes place in the strongly convex setting, and our rate is of the form $O(1/k)$ which is a fast rate for SGD and relies on *strong convexity*. Let us emphasize that obtaining such rates classically requires stronger step size conditions, see for example the following general references on stochastic optimization:
- Theorem 4.6 in *Bottou, Curtis, Nocedal (2018). Optimization methods for large-scale machine learning. SIAM review*. This features a constraint very similar to ours.
- Corollary 5.8 and Theorem 5.9 in *Garrigos, Gower (2023). Handbook of convergence theorems for (stochastic) gradient methods. arXiv preprint*. In particular Theorem 5.9 features a constraint very similar to ours.
- Moulines, Bach (2011). *Non-asymptotic analysis of stochastic approximation algorithms for machine learning. Neurips*. The discussion after Theorem 1 suggests that small step sizes are required to obtain meaningful non asymptotic bounds for SGD, as we obtain in our work.

We conjecture that the convergence of derivatives of SGD beyond the strongly convex setting is a very challenging issue. *This is not settled even for deterministic algorithms.* Our step size choice is certainly not optimal, especially in the interpolation regime, but we believe that the dependency on $\mu$ is required to obtain convergence of derivatives of SGD. This is due to the necessity to operate in the favorable strongly convex regime and the fact that this requires a possibly worse step size than the deterministic smooth case. This is aligned with the litterature on the convergence of SGD for strongly convex objectives and we will comment on these restriction and potential improvements in the revision.

## Additional experiment

One reviewer was uncomfortable with the fact that we only illustrated our findings on synthetic experiments. You will find attached on OpenRevew an additional experiment on regularised logistic regression on the ijcnn1 dataset, with the same conclusion as for the synthetic case.

---

### Decision · Program_Chairs · 2024-09-25

**Decision:**

Accept (poster)

**Comment:**

The paper looks at the behavior of the derivative of the iterates of SGD in the strongly convex setting. It has applications to bi-level optimization (through implicit differentiation). I suggest that the authors take into consideration the concerns about adding real data experiments and adding some discussions on the literature of inexact SGD and why their techniques are novel.